# Corn Silage Supplemented with Pomegranate (*Punica granatum*) and Avocado (*Persea americana*) Pulp and Seed Wastes for Improvement of Meat Characteristics in Poultry Production

**DOI:** 10.3390/molecules26195901

**Published:** 2021-09-29

**Authors:** Stefanos Leontopoulos, Prodromos Skenderidis, Konstantinos Petrotos, Ioannis Giavasis

**Affiliations:** 1Laboratory of Food and Biosystems Engineering, Department of Agrotechnology, University of Thessaly, 41110 Larissa, Greece; pskenderidis@uth.gr (P.S.); petrotos@uth.gr (K.P.); 2Laboratory of Food Microbiology, Department of Food Technology, University of Thessaly, 43100 Karditsa, Greece; igiavasis@uth.gr

**Keywords:** silage supplement, pomegranate residues, avocado residues, poultry, antioxidant, meat quality

## Abstract

In the present study, pomegranate peels, avocado peels, and seed vacuum microwave extraction solid by-products were supplemented in corn silage in order to investigate the effects on meat quality and growth rate in broiler chicken. There were 50 broilers, divided in two groups, treated with experimental or usual feed for 43 days (group A: 25 broilers fed with avocado and pomegranate by-products and group B: 25 broilers fed with corn-silage used as control). The results showed that broiler chickens fed with a diet supplemented with a mixture of pomegranate avocado by-products (group A) showed significant differences in chicken leg meat quality, significantly improving the level of proteins and fatty acids content in breast and leg meat, respectively. More specific ω3 and ω6 fatty acids content were three times higher than in group B. Moreover, a protective effect on the decomposition of polyunsaturated fatty acids, induced by free radicals and presented in chicken meat, is based on the evaluation of lipid peroxidation by measuring thiobarbituric acid reactive substances. Pomegranate peels, avocado peels, and seed by-products appeared to have a slight reduction on meat production, while it was found to improve the qualitative chicken meat characteristics. Regarding the production costs, it was calculated that the corn-silage supplementation, used in this study, lead to a 50% lower cost than the commercial corn-silage used for the breeding of broilers.

## 1. Introduction

Livestock productivity and welfare is affected by many factors including the feed used for their diet. Although feeding by-products, used as practice to feed livestock, is as old as the domestication of animals by humans [1], cereal crops such as corn, wheat, and barley are mainly used as silage for animal feed due to their high nutritional value. However, in recent years, several research efforts have exploited the possible use of novel, unconventional feedstuffs, which are rich in bioactive compounds, in silage as alternative animal feed in order to enhance livestock welfare, improve productivity, and develop new products with bio-functional activity [2,3,4]. Several wastes, such as peels and seeds of fruits and vegetables [5], bread wastes [6], sugar cane molasses [7], dairy by-products [8], dehydrated food wastes [9], raw fruit extracts [10,11], herbs [12,13], and other food waste sources [14], which are considered industrial wastes, can also be used alone or as supplements improving animal nutrition value [15,16,17]. Among food industrial wastes, avocado (*Persea americana*) and pomegranate (*Punica granatum*) are of significant importance due to their unique edible fruit [18,19,20] which contain high amounts of bioactive substances, such as polyphenols and chlorophylls with antioxidant and “radical” suppressions [21,22,23]. Although application of avocado by-products [24] and pomegranate peel in animal nutrition is still limited and not adequately documented, some research efforts have been completed in order to feed and improve animal nutrition, welfare, and meat value [25,26,27,28] in ruminants and non-ruminants [29]. More specifically, it has been found that pomegranate peel contains saponins that can improve nitrogen efficiency by reducing the activity of protozoa [27], enhancing the microbial composition of stomach proteins and increasing the protein content of the produced milk. Similar results have been commented by [30,31,32] who studied the effect of a combination of pomegranate, flaxseed seed oil, avocado, and mango fruit wastes, respectively, on the effect of daily milk production of goats, as well as blood, containing metabolites and their antioxidant status. Recent studies also investigated the effect of incorporating pomegranate by-products in the diet of lambs [33] and cows [34], while [35] commented avocado use as potential feedstuff for animals as an energy source. Besides large farm animals, bioactive substances presented in plant species and their residues also affect the health of poultry chicks [36]. So far, several research studies have focused on the harmful effects caused by oxidative stress in livestock, affecting welfare, feed efficiency, and quality of livestock products [37,38,39,40,41]. The beneficial effects caused by the use of bioactive compounds, extracted from plant residues, can lead to additional economic benefits both in animal and plant production by reducing environmental pollution and leading to the development of new and improved meat-based products. Furthermore, prohibition in the usage of meat wastes for animal feed within E.U. countries and the rising prices of fodder has lead animal breeders to alternative sources of fodder such as energy crops and fish by-products, which their usage, although limited, is not yet forbidden by E.U. regulations [42,43,44]. Furthermore, ever expanding consumer habits require meat products that are rich in natural antioxidants, without the use of synthetic additives capable of reducing the oxidation processes in meat [45]. Our hypothesis was that a mixture of avocado and pomegranate pulp and seeds could partially replace conventional ingredients in a concentrate of corn silage used for broilers production, decreasing feeding costs and improving meat characteristics. The purpose of the research study was to evaluate the use of pomegranate and avocado peel and seed residues as possible feed supplements in corn silage used for poultry growth. Broiler chicken meat production will be used as quality and quantity parameters in order to evaluate potential bioactive effects of the tested silage. In the case of quantitative criteria, the following parameters were used: (a) feed ratio to quantity of meat produced and (b) feed cost/kg of meat produced to compare the productivity and economy of silage feed with a control diet and export the relevant conclusions. The use of the food industrial wastes, as supplements for animal feed, is a sustainable approach to reducing feed cost, footprint of carbon dioxide, and producing a bio functional meat of high added value and high hygiene, performing for human health.

## 2. Results

In order to evaluate the use of pomegranate and avocado peel and seed extraction residues as possible feed supplements in corn silage used for poultry growth pH, lactic acid, and yeasts were calculated at 1st, 3rd, 5th, 12th, 19th, 26th, 33th, and 40th day after the bag was sealed. The results on pH fluctuation suggested that pH value drops from 4.6 to 4.1 until the 20th day of silage and then rises again to 4.6, probably due to the lack of sugar availability and the competition between yeasts and lactic acid bacteria that might consume organic acids. It is also important to mention that reduction in yeast and lactic acid bacteria population, at 40 days of silage, resulted in production of alkaline substances that raise the pH value. Thus, in terms of silage pH, it would be suggested to complete the fermentation between 20 and 30 days. However, lactic acid population remains consistently high for up to 40 days (Table 1).

### 2.1. Quantitative Criteria

#### 2.1.1. Broiler’s Growth Rate

Broilers weight and feed consumption were measured once a week. Thus, measurements were evaluated at day 0, 7, 14, 21, 28, 35, and 43. Figure 1 and Figure 2 represent weekly average weight/broiler and feed consumption/broiler, while Figure 3 represents average meat weight/broiler. In Figure 1, it is clear that average weight/broiler fed with basal ration (group B) was higher than the average weight/broiler fed with ration, supplemented with pulp and seeds of pomegranate and avocado by-products, every week. Although, average weight/broiler in week 1 was 65 g and 73 g for group A and group B, respectively, the difference between the two groups was higher week-by-week (Figure 3). As a result, average weight/broiler in group B, at the end of the experiment (on 43rd day), was 2.550 kg, while the average weight/broiler in group A was 2.039 kg (Figure 1). It was also observed that average weight/broiler in group A on 35th day (2.023 kg) was almost the same that observed on day 43 (2.039 kg).

Regarding average weekly feed consumption/broiler, it was observed from Figure 2 that, in the first 7 days, it consumed 0.244 kg and 0.236 kg /broiler for group A and group B, respectively. Broilers of group A consumed, in total, more feed during the experiment (122.150 kg), while broilers of group B consumed 110.720 kg. The average feed consumption/broiler was 4.886 kg for group A and 4.258 kg for group B. Comparing average broiler weight and average feed consumption, it is concluded that, while feed consumption in group A was greater than in group B, average weight of broilers in group A was lower than in group B. However, Figure 4 represents that average meat weight of slaughtered broilers did not differ statistically between the two groups (1.66 kg of meat /broiler of group A compared to 1.73 kg of meat/broiler of group B).

#### 2.1.2. Feed Cost

In order to compare the productivity and economy of silage feed with a control diet, and reach the relevant conclusions, the feed cost/kg of meat was evaluated. Then, 150 kg of conventional broilers feed was purchased from trade market and were used in this study in order to feed the broilers. The purchase cost of 150 kg of broiler feed was 0.48 Euros or 48 cents/kg. From this quantity (150 kg), 106.450 kg of corn silage was used in order to raise the broilers of group B (control), while the remaining 43.55 kg was used to be mixed with the silage to produce the feed of group B. In order to produce the feed for group B, except for the 43.55 kg of conventional broilers feed, a quantity of produced silage, in total: 78.6 kg, was added, which contained 26 kg coarsely ground maize with a price of 0.20 Euro or 20 cents. In total, the 25 broilers of group A consumed 122.150 kg of feed mixture (43.55 kg of commercial broilers feed + 78.6 kg of pomegranate-avocado silage) or 4.886 kg/broiler for group A and 4.258 kg/broiler for group B (control). Although the consumption of food in group A was 16 kg or approx. 15% higher than that consumed by the broilers of group B (control) the economic benefit resulting from the consumption of food for group A is much greater than that of group B, due to the difference in prices of the two feeds. In more details and at economic terms, the feed cost for the 25 broilers of group A was 43.55 kg × 0.48 Euros/kg + 26 kg ground maize × 0.2 Euros/kg = 26.1 Euros while the feed cost for the broilers of group B was 106.15 kg × 0.48 Euros/kg = 50.95 Euros. Therefore, the feeding cost of control group B was 1.95 -fold higher than that the feeding cost of group A although the feed consumption was higher.

### 2.2. Qualitive Criteria

#### 2.2.1. Fat, Protein Moisture and Collagen Measurements

Dietary manipulation is an important way to improve meat quality in poultry. Broiler meat concentration in fat, protein, collagen, and water holding capacities (moisture) are associated with meat quality characteristics [46]. The effect of the tested dietary on meat quality indicators are given in the Table 2.

Fat, protein, moisture, and collagen of each group (A and B) were measured using a NIR device. The results presented in Table 2 show that the fat content was higher in broiler leg tissues than in breast in both groups. Regarding protein content, in both groups, protein content found in breast was higher than protein content found in chicken leg, varied from 22.57–22.86 and 19.76–20.04, respectively. Protein content in leg meat of broilers, fed with pulp and seeds of pomegranate and avocado by-products, was slightly higher (0.28) than those observed in broilers of group B. Moisture of breast and leg meat in both groups (A and B) was not different statistically and it was varied between 74.96–75.21%. Finally, collagen in breast meat of broilers of group A was lower (0.79) than collagen content (0.88) found in breast meat of chickens fed with a basal content (group B). However, collagen content was slightly higher (0.01) in leg meat in group A, compared with collagen found in leg meat of group B. In all cases, fat, protein, and collagen contents in chicken meat were higher in chicken legs than in chicken breast.

The analysis of the samples were performed, due to the small sample size and the normality of the sample values, with the *t*-test and Mann–Whitney Test. Based on the results, an important difference among protein content in chicken leg muscles were observed. From the *t*-test analysis, it was found that there is a statistically important difference in protein content observed in muscle of chicken legs based on the *p* = 0.028. Additionally, according to the Mann–Whitney Test, there is a statistically important difference (*p* = 0.0220) in lipid contents found in chicken breast muscles.

#### 2.2.2. Thiobarbituric Acid–Reactive Substances (TBARS) Assay

Although thiobarbituric acid–reactive substances (TBARS) assay has been criticized, mainly due to its potential lack of specificity, caused by the harsh conditions during sample preparation, and the possible cross reactivity of TBA with other aldehydes is still remaining one of the most popular assays for lipid peroxidation [47]. In the present study, it was found that oxidative stress biomarker TBARS, measured in leg and breast muscle, showed that corn silage, supplemented with pulp and seeds of pomegranate and avocado by-products, improved the redox status of the broiler chickens by significantly deceasing TBARS (mg/kg MDA) in group A compared to control group B, as it is shown in Table 2. The group containing avocado and pomegranate feed was the most potent, exhibiting the greatest decrease in TBARS levels.

#### 2.2.3. Lipid and Fatty Acid Profile

For determination of omega 3, omega-6/omega 3 ratio, monounsaturated and polyunsaturated fatty acids, a sample of small amount of homogenized five legs, and a sample of homogenized of five breasts were used. From Table 3, it was observed that saturated fats in chicken breast fed with avocado, pomegranate, and corn silage was 31.86% (group A), while in samples of group B (control), it was 31.45%. Furthermore, monosaturated and polysaturated fats were 50.44%, 48.57%, 17.69%, and 19.0,2 respectively. However, percentage of total ω3 lipid acids observed in chicken breast of group A (fed with avocado, pomegranate, corn silage) was three times higher compared to the control group B (1.19% and 0.41%, respectively). Similar results were observed for total trans (1.08% and 0.27%, respectively), while total ω6 lipid acids for group A was 16.5% and group B was 18.6%. Similar results were observed for chicken leg samples. More specifically, saturated fats in chicken leg fed with avocado, pomegranate, and corn silage was 29.46% (group A), while in samples of group B (control), it was 31.09%. Furthermore, monosaturated and polysaturated fats were 51.27%, 48.88%, 19.24%, and 19.06%, respectively. However, percentage of total ω3 lipid acids, observed in chicken leg of group A (fed with avocado, pomegranate, corn silage), was three times higher compared to the control group B (1.33% and 0.41%, respectively). Similar results were observed for trans (1.08% in group A and 0.27% in control group B) while total ω6 lipid acids for group A was 16.5% and group B was 18.6%, respectively. Finally, comparing total ω3 contents between the two different tissues (chicken breast and chicken leg), it was observed that ω3 contents in chicken legs was higher (1.33%) than ω3 contents found in chicken breast (1.19%). The ratios of PUFA/SFA and ω6/ω3 are significant for human health status and, as recommended by the World Health Organization [48], the PUFA/SFA ratio should be above 0.4–0.5.

## 3. Discussion

Important factor for animal production is not only the nutrition value of many forage species but also the maintenance of their health condition. Large scale poultry production is characterized by provision of veterinary antibiotics for either treatment or prevention of infectious diseases or to increase their productivity [49]. Reducing and replacing feed additives and veterinary antibiotics, with natural compounds such as plant extracts and products originated from agricultural wastes, have become a global challenge, and their alternative use for various purposes is currently of great importance [50,51,52,53,54,55]. Antimicrobial properties, such as prevention in bacterial colonization, enzyme inhibition, cell membrane breakdown, and substrate deprivation are among positive effects that have resulted from the use of phenolic compounds [56] and thus the administration of natural antioxidants has been suggested for protecting farm animals from such pathologies or mitigating their symptoms [57,58]. Most of the antimicrobial studies focus on the toxicological and nutritional properties of animal feed and overlook the possible effects of this particular diet on animal growth rate and meat quality. This purpose is achieved by preventing or minimizing lipid oxidation. So far, it is well known that the use of natural antioxidants such as ascorbic acid, tocopherols, and phenolic compounds found on animal meat products affect oxidative stability of muscle proteins [59,60,61], improving and extending the life of meat [62,63,64] and milk yield and quality [65].

The findings of the present study suggested that the incorporation of pomegranate and avocado byproducts into the feed improved the chickens’ health status and wellness, while productivity of fresh meat was not seriously affected. These results are in accordance with study completed by [2] where feed supplemented with grape pomace decreased oxidative stress-induced toxic effects.

Regarding the use of avocado residues, avocado oil, and phenol-rich extracts from the bark and seeds have been highlighted as inhibitors of oxidation of lipids and proteins [22,23]. Furthermore, Rodríguez-Carpena et al. [66] and Utrera et al. [67] investigated the beneficial effects of avocado residues on the nutritional value in processed pork meat. In particular, lipid and protein cravings are resulted by the significant increase in monounsaturated fatty acids (MUFAs), tocopherols, flavonoids, and chlorophylls. In addition, it has been commented that avocado residues obtained from two commercial varieties (“Fuerte” and “Hass”) significantly reduced redness loss of stored pork meat concluding that they can be used as ingredients for the production of foods with enhanced quality characteristics [22]. More specifically, “Fuerte” variety was more effective in preventing the discoloration of the frozen meat than the extracts of the “Hass” variety, while avocado “Hass” variety extracts, significantly inhibited the formation of carbonyl proteins in a cooling medium after 15 days of storage. According to Naht Dinh [68], palmitic and oleic acid are the most important FAs, not only in subcutaneous fat, but also in important muscles such as the *Longissimus lumborum* et *thoracis*. Since, linoleic and -linolenic are the main -6 and -3 PUFAs, respectively, and are important for human nutrition and health [69], their increase in chicken meat reveals the beneficial effect of inclusion of avocado and pomegranate addition in the feed in poultry diet on broiler quality. PUFA/SFA and (MUFA + PUFA)/SFA ratios did not differ between treatments.

Hernández-López et al. [70] also studied the use of avocado wastes as feed additives in a mixed diet to feed piglets in order to contribute to sustainable agriculture production and reduction of pork-meat production cost. In their study, muscle tissues were analyzed for composition, oxidative and color stability and compared with muscle tissues obtained from pigs fed with a diet that did not contain avocado wastes. Feeding pigs with avocado residues had a significant effect on the content and composition of intramuscular fat, reducing lipid content in muscles tissues, increasing the degree of satiety. They also concluded that muscles tissues had significantly lower rates of lipid and protein oxidation during refrigeration.

Regarding pomegranate, Abdel Aziz et al. [36] compared in vitro the anthelmintic efficacy of different concentrations (25, 50, and 75 mg/mL supplemented with phenbendazole at a concentration of 5 mg/mL) of ethanolic pumpkin extract and pomegranate peel against *Ascaridia galli* worms’ infection in Baladi chicks. Results of this study showed that pomegranate extract had a lower lethal effect than phenbendazole and anthelmintic efficacy is depending on the time and concentration of application.

So far, the use of natural antioxidant compounds has been proposed by various researchers in order to reduce the oxidative stress [71,72]. Studies regarding the reduction in TBARS levels have shown that some of the major polyphenolics are effective to inhibit lipid peroxidation [73,74]. Thus, it is believed that the use of natural antioxidants could reduce lipid peroxidation and enhance quality of chicken meat [75] decreasing nutritional values of meat and meat products [76]. Furthermore, according to Naveena et al., (2008) [77] Pomegranate rind powder phenolics significantly inhibited lipid oxidation in cooked chicken patties to a much greater extent than pomegranate juice and butylated hydroxyl toluene (BHT). Additionally, 10 mg equivalent phenolics/100 g meat would be sufficient to protect chicken patties against oxidative rancidity for periods longer than BHT. The meat industry could use these fruits or fruit byproducts as a potential source of phenolics as they have immense nutraceutical value and can be used to produce functional meat products of commercial interest.

Despite pomegranate wastes, Gerasopoulos et al. [78] commented that chicken redox status was improved due to the incorporation of antioxidant polyphenols from by-products of OMWW to chickens’ broiler feed. Furthermore, Mujahid et al. [79] also mentioned the administration of olive oil-supplemented diet as natural antioxidant offering a skeletal muscle protection from heat stress-induced oxidative stress. It is also believed that 30 days of experimental feed is adequate time to profound a significant increase in total antioxidant capacity (TAC) in plasma as well as in quality characteristics in meat products.

Beside studies in chicken, pomegranate residue has been tested for their potential positive effect in diet of several livestock animals. Oliveira et al. [80] commented that a diet enriched with pomegranate extract affects fat burning system and raw protein in young calves during the first 70 days of their breeding possibly due to high levels of tannins. Jami et al. [81] and Shabtay et al. [82] also commented the positive effect of a diet enriched with 4% pomegranate peel extract in significant increase in dry digestibility, raw protein, fiber and milk production in calves and cows. Khorsandi et al. [34] also studied the effect of pomegranate by-products silage under heat stress condition in nutrients digestibility, metabolic parameters and milk production in postpartum Holstein cows. In their study it was concluded that replacing CS with 120 g PBS milk yield in Holstein dairy cows under thermal stress appear higher ADF digestibility and lower protein (CP) digestibility without any adverse effect on blood metabolites or their health. A similar study was conducted by Safari et al. [83] who assessed whether pomegranate seeds or the combination of peel and seed, can provide effective action to improve the condition of antioxidants and therefore the metabolic profile and yield in Holstein dairy cows.

Emami et al. [30] investigated the effect of extracts of pomegranate and flaxseed oil on dry matter intake, protein, lactose and milk yield of goats. In their study they concluded that antioxidant status and milk fat of dairy goats may be possible be improved by adding 25 g/kg DM of pomegranate seed oil to the diet of animals, more specific, the results of this study showed that production efficiency in dietary treatments was similar in different groups, but milk fat increased with the addition of flaxseed and pomegranate seed oils. In terms of blood contains, metabolites (glucose, total cholesterol, triglycerides, LDH and LDL) were not affected by dietary therapies. Compared to the diet with flaxseed oil, the use of pomegranate oil extract increased total antioxidant capacity.

Natalello et al. [84] investigated the use of pomegranate by-product as feed supplement in order to feed lambs. More specific, in their study 17 lambs were fed with 200 g/kg cereal-based concentrate of pomegranate by-product. Their results shown that α- and γ-tocopherols (vitamin E), polyunsaturated fatty acids (PUFA) and a peroxidase index were improved in lambs fed with pomegranate by-product supplement. Furthermore, formation of methymoblobin was reduced after 7 or 4 days for cooked meat while meat antioxidant properties were improved in lipophilic fraction, but not in the hydrophilic. Furthermore, Andrés et al. [85] in their study observed an increase in the concentration of linoleic acid, trans-10, co-merged linoleic acid and total phenol content in lamb meat when they fed with 240 g/kg DM resulted an improvement of quality characteristics of lamb meat products. Similar results were observed form a study completed by Kotsampasi et al. [86] where the use of pomegranate by-products in a diet as a food supplement affected positively quality characteristics such as fat color, fat stability, moisture and overall efficiency. However, lamb meat quality was not affected by the use of pomegranate by-products in a diet as a food supplement.

Beside farm animals pomegranate residue extracts antioxidant effect have been studied in aquaculture. Wu et al. [28] studied the use of pomegranate residue as a natural antimicrobial in order to improve resistance to fish disease. From their findings it was concluded that the number of pathogenic bacteria and the cumulative mortality of Darkbarbel catfish that received a diet with pomegranate residues decreased by 15–45% compared to the untreated control group.

Productivity and economy of silage feed is also an important key factor for farm business. Several by-products of agricultural processing (e.g., fruit pulps and fruit residues resulting from juicing procedure) appear increased management costs and/or can pollute the environment when disposed-off without any treatment. However, these by-products often hide a small “treasure”, i.e., natural antioxidants-antimicrobials-bioactive substances [87,88,89] that can be isolated and utilized from the waste, providing high added value in a product and affecting health and productivity of farm animals but also designate the cost of livestock products. The aforementioned approach is also an EU guideline on bio-refinery. This approach also leads to a reduction in the carbon footprint of solid waste due to their total utilization and therefore their re-introduction into the food chain. In addition, the synergy of biologically produced substances can further increase their activity, especially in microbiostatic or microbicidal action according to the well-known theory of multiple barriers [90]. Therefore, the result of the optimized isolation with innovative “green” extraction method such as vacuum microwave assistance has been completed in previous studies [91,92,93] this research is focused in the use of these residues as animal feed. The proposed total utilization of the three solid residues by the food industry (Total Discharge approach) is a pioneer and can provide high value-added materials with low raw material costs. Regarding the production costs of these feeds, it is stated that the silage used in this study was 75% cheaper than the commercial silage used for the breeding of broilers. Thus, the cost reduction and the nutritional and the environmental benefits, regarding the use of these wastes, are believed to be significant advantages for their acceptance.

## 4. Materials and Methods

### 4.1. Plant Materials

Avocado and pomegranate peel and seed by-products were collected as by-products after Vacuum Microwave-Assisted Extraction. The extraction was performed on Pella’s Nature P.Co. located in Macedonia, Greece, facilities using an industrial scale Vacuum Microwave-Assisted Extractor. The obtained by-products were kept in a freezer at −20 °C until further use as supplements in corn silage.

### 4.2. Preparation of the Supplemented Silage

Pomegranate and avocado fruit by-products (pulp and seeds) were added in broiler feed as corn silage. For this reason, coarsely ground maize grains was mixed with pomegranate pulp, avocado pulp and avocado seed wastes. The silage corn contained 35% solids (23.5% grounded corn seeds and 76.5% pomegranate-avocado pulp and seed wastes mixed) and 65% moisture. The moisture of the pomegranate pulp, avocado pulp, and avocado seeds wastes were calculated before application according to the method described by [94] and it was 88%, 81% and 77% respectively (average moisture of the three wastes 82%). Moisture of grounded corn seeds was 12%.

A concrete type mixer was used in order to mix the grounded corn and the pomegranate-avocado by-products. A standard commercial formulation of *Lactobacillus*
*buchneri* lactic acid bacteria (Pioneer silage inoculant 11A44) was used for the lactic fermentation of corn and the preparation of the corn silage. Additionally, 10 g of lactic acid bacteria had been dissolved in water (10% *w*/*v*) by stirring prior to mixing with 100 kg of corn-avocado and pomegranate pulp and seed wastes. For corn silage production supplemented with avocado and pomegranate pulp and seed wastes the final mixture was placed in special air tight-seal plastic bags and was fermented for about 40 days [95]. In order to prevent sealed bags from rupturing due to the inflation caused by the carbon dioxide production during fermentation, extra gas was absorbed using a vacuum cleaner every day. pH, lactic acid, yeasts and total lactic acidity were calculated at 1st, 3rd, 5th, 12th, 19th, 26th, 33th, and 40th day after the bag was sealed. The duration of silage was 43 days with an expected drop in pH in the range of 3.5–4.0. At the end of the silage, the critical parameters for the quality of the silage produced was measured and evaluated. The quality parameters evaluated in this study was: (i) total crude protein content %, (ii) organic acid profile, and particularly fatty acid %, total ω3 fatty acids, total ω6 fatty acids, ω3/ω6 ratio, saturated fats, monounsaturated fats, polyunsaturated fats, protein content, moisture, and collagen content

### 4.3. Broilers

Fifty broiler chickens (seven days of age) were used in this study in order to evaluate corn silage supplemented with pomegranate and avocado fruit by-products (pulp and seed wastes). Race “Ross’ broilers were purchased from the “Agrafiotis” local aviary (Tyrnavos, Greece) and grown on facilities of University of Thessaly farm. Broilers were housed under controlled environmental conditions (12-h light/dark cycle, temperature 25–33 °C, humidity 50–70%) in standard facilities for each group. The chickens were randomly divided into two experimental groups (25 broilers per group) as follows: a) a group fed with ration supplemented with pulp and seeds of pomegranate and avocado by-products and b) control group fed with basal ration. The waste and by-products of avocado and pomegranate usually have high-moisture content, and are fibrous, Thus, their caloric value is negligible and the caloric value of feed content for group A and group B was 3.05 Mj/728.46 Kcal and 12.2 Mj/2.91 Kcal respectively (Table 4).

During their growth water and different ration were provided *ad libitum* for 43 days. Broilers were also vaccinated for bronchitis (IB) (*Avian coronavirus*), Gumboro disease known also as infectious bursitis, infectious avian nephrosis, and infectious bursal disease (IBD) and Avian pneumoencephalitis (Newcastle disease-ND) (*Avian orthoavulavirus*) after 7 (for IB and IBD) and 14 days, respectively.

Once a week, weight of the used ration and the grown broiler was measured. After 43 days, broilers were led to the slaughter house and body weight and tissue samples were collected in order to determine quantity and quality characteristics. For tissue collection and dead body weight, broilers were sacrificed in a fully automated slaughter complex in “Agrafiotis” local aviary company. Skin removal, viscera separation, slaughter, bleeding, washing, and other relevant procedures were conducted by specialized staff used special equipment. The experiment was reviewed and approved by institutional review board of the University of Thessaly.

### 4.4. Tissue Collection

Broilers sacrificed in a fully automated slaughter facility in order to collect tissues for further analysis. There were 200 g breast muscle samples and 5 leg muscle samples collected from each group. Each tissue sample was then grinded using a minced meat machine and measured in NIR device (Figure 5) in order to determine protein, fat, moisture, and collagen contents. Collected tissues from leg and from the breast of 5 sacrificed broilers were used in order to determine antioxidants using Thiobarbituric acid reactive substances for TBARS assay. In addition, fatty acid profiles were measured for both diet groups with emphasis on omega-3 fatty acid content, omega-6/omega-3 ratio as well as monounsaturated and polyunsaturated fatty acids. Each assay was performed in triplicate.

### 4.5. TBARS Assay

Thiobarbituric acid–reactive substances (TBARS) were determined using a slightly modified assay of Keles et al. [96] by Gerasopoulos et al. [97]. In more details, 50 μL of leg or breast muscle sample homogenized was mixed with 500 μL of 35% TCA and 500 μL of Tris-HCl (200 mmol/L; pH 7.4) and incubated for 10 min at room temperature. One milliliter of 2 mol/L Na_2_SO_4_ and 55 mmol/L thiobarbituric acid (TBA) solution was added, and the samples were incubated at 95 °C for 45 min. The samples were cooled on ice for 5 min and were vortexed after 1 mL of 70% TCA was added. The samples were centrifuged at 15,000× *g* for 3 min, and the absorbance of the supernatant was read at 530 nm. A baseline shift in absorbance was taken into account by running a blank along with all samples during the measurement. Calculation of TBARS concentration was based on the molar extinction coefficient of malondialdehyde (MDA) equivalents. MDA forms a 1:2 adduct with TBA, which can be measured by spectrophotometry. The molar coefficient of MDA is 155 × 10^3^ M^−1^cm^−1^. The assay was performed in triplicate.

### 4.6. Lipid Extraction and Fatty Acid Analysis

Each poultry meat sample was homogenized and total lipids were extracted using the Folch method [98]. In particular, 1 g of each homogenized sample was mixed with 20 mL chloroform:methanol (2:1, *v*/*v*). The mixture was vortexed and water was added for the phase separation. The upper phase was removed and the lower one was collected. After solvent evaporation, 0.1 g of the extracted lipids were weighed in a test tube with a screw cap. Afterwards, 2 mL of heptane were added, followed by 0.2 mL of a 2 M methanolic solution of potassium hydroxide for the fatty acid methyl esters (FAME) preparation. The mixture was vortexed for 30 s and was left to settle till the upper phase became transparent. Then, this phase that contains the methyl esters was collected and analyzed by gas chromatography with flame ionization detector (Agilent 8890 GC System) equipped with an autosampler (Agilent 7693A System). FAME were analyzed on a HP-88 column (100 m × 0.25 mm; 0.20 μm, J & W Scientific, Agilent Technologies). Hydrogen was the carrier gas at a flow rate of 0.928 mL/min. The injector port and detector temperature were maintained at 260 °C. The column oven was programmed to maintain a temperature of 100 °C for 5 min, then rise to 180 °C at 8 °C/min and maintain that temperature for 9 min, before rising to a plateau of 230 °C at a rate of 1 °C/min for 15 min. The total run time was 89 min. FAME were identified by comparing retention times to a standard mixture containing 37 fatty acids.

### 4.7. Statistical Analysis

ANOVA statistical analysis was used in order to evaluate the use of pomegranate and avocado peel and seed residues as possible feed supplements in corn silage used for poultry growth. The level of statistical significance was set at *p* < 0.05. All results are expressed as mean ± SEM. Data was analyzed using Minitab 17). To perform the statistical analysis using the *t*-test, the following steps were followed due to the small number of samples (*n* < 30): Using Minitab statistical package data normality was checked for both comparative samples. If the *p*-value was > 0.05 then a *t*-test was performed. If from the initial test of the normality of the data, the *p*-value was <0.05 for both comparative samples, the non-parametric “Mann–Whitney Test” was used.

## 5. Conclusions

In the context of the present research work, functional silage was produced by mixing, and consequently fermenting, by commercial lactic acid bacteria culture coarsely ground maize seeds in combination with post extraction residuals of pomegranate peels and avocado solid waste. Then, the produced silage was incorporated at high level in conventional broiler feed in order to produce a novel functional feed for broiler feed for meat production. The produced novel feed was used, experimentally, in broilers nutrition in comparison to a conventional feed used as control. The experimental results indicated that the novel bioactive feed performed better than the conventional one because it led to the production of chicken meat of high quality, with less feed cost per kilo of produced meat and very rich in mono-saturated fatty acids, as well as ω-3 fatty acids and in addition with a reduced ratio of ω6/ω3. Furthermore, the proposed inclusion of the solid wastes of the pomegranate and avocado industries in the broilers nutrition, in the form of silage, results in a sustainable disposal of these solid wastes, which contributes to the reduction of the carbon footprint of the involved industries. Therefore, the novelty of the present research work is founded on the sustainable and mass disposal of the solid wastes of the pomegranate and avocado industries and on the production of high added value bioactive chicken meat. Finally, this high added value bioactive chicken meat is produced with substantially lower cost, due to the high percentage incorporation of the zero cost solid wastes in the broiler feed. As a result of this research work, the development of a patented methodology, regarding bioactive silage, produced by a mixture of solid wastes of pomegranate and avocado pulp and seeds (patent application number: 245-0004293647 deposited at Hellenic Industrial Property Organization).

## Figures and Tables

**Figure 1 molecules-26-05901-f001:**
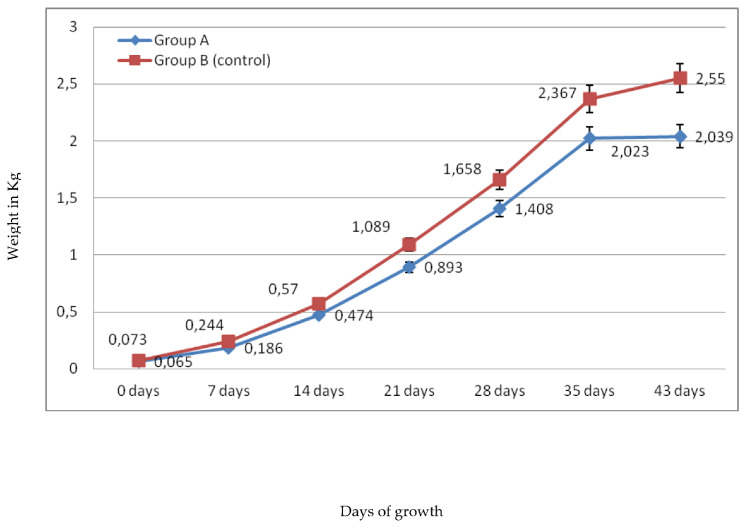
Broiler average weight after 43 days of growth. Broilers of group A, fed with ration supplemented with pulp and seeds of pomegranate and avocado by-products, and broilers of group B (control group), fed with basal ration.

**Figure 2 molecules-26-05901-f002:**
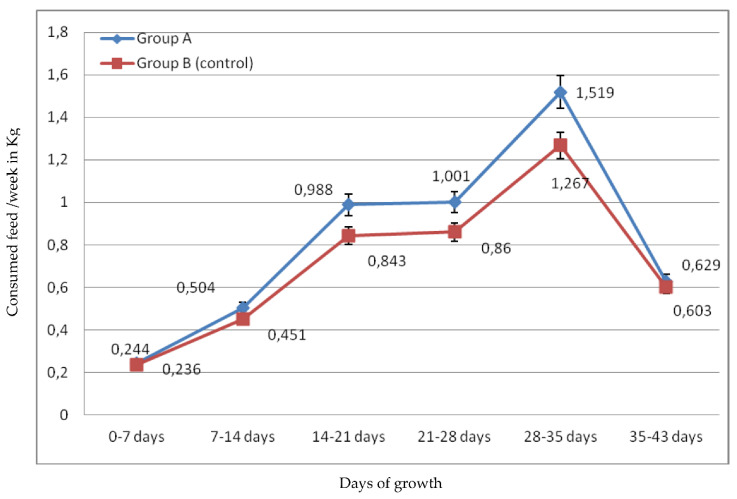
Broiler average feed consumption/week. Broilers of group A, fed with ration supplemented with pulp and seeds of pomegranate and avocado by-products, and broilers of group B (control group), fed with basal ration.

**Figure 3 molecules-26-05901-f003:**
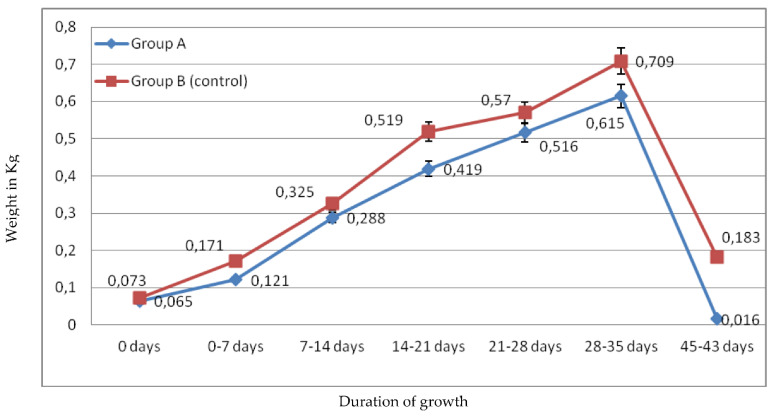
Broiler average growth rate/week. Broilers of group A, fed with ration supplemented with pulp and seeds of pomegranate and avocado by-products, and broilers of group B (control group), fed with basal ration.

**Figure 4 molecules-26-05901-f004:**
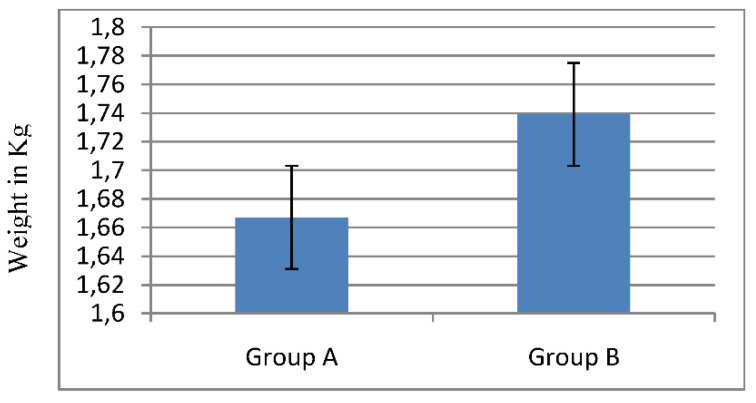
Average meat weight of sacrificed broilers. Broilers of group A, fed with ration supplemented with pulp and seeds of pomegranate and avocado by-products, and broilers of group B (control group), fed with basal ration.

**Figure 5 molecules-26-05901-f005:**
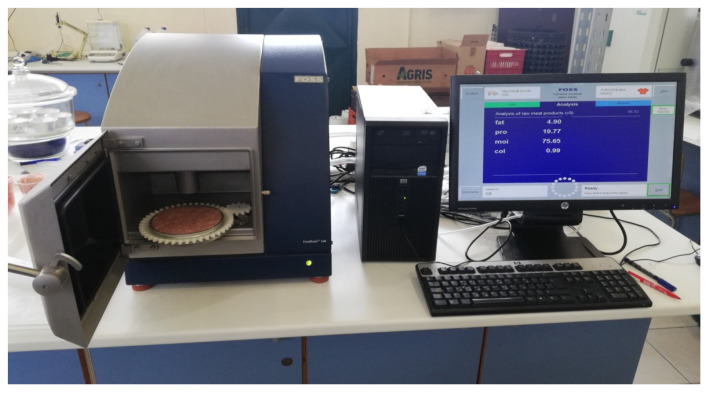
NIR device.

**Table 1 molecules-26-05901-t001:** Effect of silage time on growth of pH, lactic acid, and yeast population (log cfu/g) in corn silage, supplemented with pomegranate and avocado peel, used as feed for poultry growth.

Days	pH	Lactic Acid Bacteria Log cfu/g	Yeasts Log cfu/g
1	4.56 ± 0.03	7.35 ± 0.04	6.08 ± 0.08
3	4.50 ± 0.01	7.62 ± 0.05	7.36 ± 0.06
5	4.48 ± 0.02	8.79 ± 0.10	7.95 ± 0.05
12	4.27 ± 0.01	8.93 ± 0.04	8.56 ± 0.07
19	4.16 ± 0.01	8.88 ± 0.07	8.52 ± 0.08
26	4.42 ± 0.01	9.09 ± 0.06	8.70 ± 0.1
32	4.53 ± 0.02	9.02 ± 0.07	8.58 ± 0.11
40	4.59 ± 0.01	8.84 ± 0.04	8.18 ± 0.08

Measured values are medians of three repetitions ± standard deviation.

**Table 2 molecules-26-05901-t002:** Effects on oxidative stress markers (TBARS assay).

	Broiler’s Tissue Muscle	TBARS mg/kg MDA
Group A (Avocado + pomegranate)	Leg	0.076 ± 0.2 *
	Breast	0.076 ± 0.2
Group B (Control)	Leg	0.149 ± 0.3
	Breast	0.080 ± 0.2

Group A: Broilers fed with corn silage, supplemented with pulp and seeds of pomegranate and avocado by-products. Group B: Broilers fed with commercial corn silage, used as control. Measurements that have a statistically significant difference to the control are highlighted by an asterisk (*). Measured values are medians of three repetitions ± standard deviation. The level of significance was set at *p* ≤ 0.05.

**Table 3 molecules-26-05901-t003:** Fatty acid content and nutritional ratios, observed in breast and leg of broilers fed with corn silage supplemented with avocado and pomegranate (group A) or commercial silage (group B-control).

Fatty Acid %				
	Group A (Avocado + Pomegranate + Corn)	Group B (Control: Corn Sillage)	Group A (Avocado + Pomegranate + Corn)	Group B (Control: Corn Sillage)
	(Chicken Breast)	(Chicken Breast)	(Chicken Leg)	(Chicken Leg)
Myristic (C14:0)	0.49 ± 0.0	0.55 ± 0.0	0.50 ± 0.0	0.53 ± 0.0
Myristoleic (C14:1 cis-9)	0.14 ± 0.0	0.16 ± 0.0	0.15 ± 0.0	0.16 ± 0.0
Pentadecanoic (C15:0)	0.08 ± 0.0	0.09 ± 0.0	0.08 ± 0.0	0.09 ± 0.0
Palmitic (C16:0)	24.99 ± 0.1	24.45 ± 0.2	22.83 ± 0.2	23.61 ± 0.2
Palmitoleic cis (C16:1 cis-9)	5.97 ± 0.1	5.17 ± 0.1	6.05 ± 0.1	5.54 ± 0.1
Stearic (C18:0)	6.19 ± 0.0	6.35 ± 0.1	5.84 ± 0.1	6.63 ± 0.1
Oleic (C18:1 cis ω9)	44.32 ± 0.2	43.27 ± 0.2	45.07 ± 0.2	43.18 ± 0.2
Linoleic (C18:2 cis ω6)	16.27 ± 0.1	18.44 ± 0.2	17.65 ± 0.2	18.37 ± 0.1
Linolenic trans (C18:3 ω3)	1.08 ± 0.0	0.27 ± 0.0	1.18 ± 0.0	0.27 ± 0.0
g-Linolenic cis (C18:3 cis ω6)	0.23 ± 0.0	0.17 ± 0.0	0.25 ± 0.0	0.29 ± 0.0
Eicosatrienic (C20:3 cis ω6)	0.11 ± 0.0	0.13 ± 0.0	0.15 ± 0.0	0.14 ± 0.0
Eicositrienoic (C23:0)	0.11 ± 0.0	<0.01	0.20 ± 0.0	0.23 ± 0.0
SFA	31.86	31.45	29.46	31.09
MUFA	50.44	48.57	51.27	48.88
PUFA	17.69	19.02	19.24	19.06
(PUFA+MUFA)/SFA	2.14	2.15	2.39	2.19
PUFA/SFA	0.56	0.60	0.65	0.61
Total ω3 fatty acids	1.19	0.41	1.33	0.41
Total ω6 fatty acids	16.5	18.6	17.91	18.65
ω6/ω3	13.87	45.37	13.47	45.49
Fat (%)	1.99 ± 0.04 ^b^	1.79 ± 0.03 ^a^	5.19 ± 0.03 ^A^	5.09 ± 0.03 ^A^
Protein (%)	22.57 ± 0.3 ^a^	22.86 ± 0.4 ^a^	20.04 ± 0.03 ^B^	19.76 ± 0.03 ^A^
Moisture (%)	75.07 ± 1.13 ^a^	75.21 ± 0.98 ^a^	74.96 ± 0.03 ^A^	75.18 ± 0.03 ^A^
Collagen (%)	0.79 ± 0.08 ^a^	0.88 ± 0.03 ^a^	1.07 ± 0.03 ^A^	1.06 ± 0.03 ^A^

The reported values are mol.% mean values ± SD (*n* = 3), calculated using peak area values corrected with theoretical response factors. SFA: saturated fatty acid; MUFA: monounsaturated fatty acid; PUFA: polyunsaturated fatty acid. Measured values of fat, protein, moisture, and collagen are medians of three repetitions ± standard deviation. Different letters indicate differences in the means within each raw of the same tissue column per each group estimated by *t*-test and Mann–Whitney Test pair wise comparisons. Means that do not share a letter are significantly different at *p* ≤ 0.05.

**Table 4 molecules-26-05901-t004:** Ingredients and nutrient composition of basal feed (Group B) and on corn silage supplemented with pomegranate and avocado pulp and seed wastes (Group A).

Compounds	Group A	Group B
Total proteins *	16.38	18.90
Total fat *	4.43	4.69
Total fibre *	2.61	2.34
Total ash *	4.23	4.41
Carbohydrates *	59.83	55.42
Sugars *	2.4	2.2
Moisture *	10.35	10.80
Ca *	0.25	0.9
P *	0.13	0.46
Available P *	0.32	0.41
Na *	0.14	0.16
Methionine *	0.40	0.43
Lycine *	0.93	1,08
Energy per kg	3.05 Mj/728.46 Kcal	12.2 Mj/2.91 Kcal

* Percentage %.

## Data Availability

Not applicable.

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
