# Peer review of "Corn Silage Supplemented with Pomegranate (Punica granatum) and Avocado (Persea americana) Pulp and Seed Wastes for Improvement of Meat Characteristics in Poultry Production"

_molecules, 2021, doi:10.3390/molecules26195901_

Round 1

Reviewer 1 Report

Dear Authors,

your research is interesting, but in this form (in my opinion) it cannot be published. Much information is missing, some are unclear

  • The introduction is poorly written
  • What was the caloric value of feed, protein content (gr A and B)? Maybe the feed A was too low in calories since the consumption increased and the weight gain decreased?
  • No research on biochemical indicators - you cannot talk about bird health
  • No tests of the intestinal microflora
    the discussion does not include the production results and meat quality (proteins, fatty acids)
  • Other comments are included in the text of manuscript

Author Response

Dear reviewer,

First of all, we would like to thank you for the efforts you made to improve this manuscript. We made all the proposed corrections. Please check our responses in the observations.

REVIEWER1

Reviewer 1  Comments

Corrections

Generally, your research is interesting, but in this form (in my opinion) it cannot be published. Much information is missing, some are unclear

The manuscript has been significantly improved and we believe that now is meet the criteria for publication.

The introduction is poorly written

The introduction has been rewritten and now we believe that is more specific.

What was the caloric value of feed, protein content (gr A and B)? Maybe the feed A was too low in calories since the consumption increased and the weight gain decreased?

We agree with the reviewer that is good to put the figures about the caloric values in the paper but the lower weight in the bioactive treatment compared to the control is not a problem as the ratio of cost of the feed vs Kg of produced meat is due to the incorporation of the solid wastes much in favor of the bioactive treatment and this is what really matters for the farmers.

No research on biochemical indicators - you cannot talk about bird health.

The purpose of the research work was not the examination of bird health but the benefits of the antioxidant bioactive substances presented in both tested fruit wastes used as feed supplements on broiler’s meat. Moreover, there was no loss of broilers during the experiments from both groups. Furthermore, oxidative stress is higher when a broiler has not good welfare. Thus, there are data presented in the paper that give information about the oxidative status of the animals of the control group and of bioactive feed group expressed as measurement of the TBARS. There are also data on the fatty acid composition in the fat of the two respective groups and this are significant markers to prove less oxidative stress in the broiler group fed with the waste silage compared to the control group fed by conventional feed.

No tests of the intestinal microflora the discussion does not include the production results and meat quality (proteins, fatty acids).

The purpose of the research work was not the examination of the microflora. However, macronutrients(meat proteins and lipds) as well as the fatty acids profile analyses are included and commented in the paper.

Line 493   one day old?

7 days old

Line 498   What about subgroups?

There was no need to subgroup the broilers

Table 7   Is it calculation or tested? Where is group A's data?

Data of table 4 (formerly table 7) are collected from the technical specifications of the feed supplier. Data of Group B have been also included in table 4.

Line 523   Why only 5 samples?

The samples were homogenized from all broilers. 5 samples were written by mistake.

Some other minor corrections (like. and ,) have been included to the text.

Reviewer 2 Report

General comments

This study focused on the effects of pomegranate peels and avocado peels and seeds supplemented in corn silage on meat quality and growth rate in broiler chicken. This research is interesting.

However, there is no hypothesis and no clear objective. Why you use pomegranate peels and avocado peels and seeds as supplements in corn silage?

The Introduction is too too long, please be more concise and specific. On the other hand, the manuscript is not written well. The Figures need to be modified to be more clear and beautiful. The Table should compare Group A and Group B. Table 2 and Table 6 should be consistent. Where are Table 3 and Table 4.

Line-by-line comments

Line 32-43: 9-16? Too many references for this expression. Suggest being more specific.

Conclusions

Please revise the conclusion. It is too long. You should focus on the main finding of this study.

Author Response

COVER LETTER

Dear reviewer,

First of all, we would like to thank you for the efforts you made to improve this manuscript. We made all the proposed corrections. Please check our responses in the observations.

REVIEWER  2

Reviewer 2  Comments

Corrections

However, there is no hypothesis and no clear objective. Why you use pomegranate peels and avocado peels and seeds as supplements in corn silage?

Hypothesis and clear objective of the research work has been added to the manuscript at the end of the introduction. In more details, the purpose of the presented research work is that improving oxidative stress, meat quality characteristics also improved since the meat supplemented with the avocado and pomegranate pulp and seed wastes contained increased ω3 fatty acids and low rate of ω6/ ω3 fatty acids and can be said that the meat produced is a biofunctional meat. The use of the food industrial wastes as supplements for animal feed is a sustainable approach reducing feed cost, footprint of carbon dioxide, producing with a sustainable way bio functional meat of high added value and high hygiene perform for human health.

The Introduction is too long, please be more concise and specific

The introduction has been rewritten and now we believe that is more specific and not too long.

The Figures need to be modified to be more clear and beautiful.

Resolution of figures have beenmodified and improved

The Table should compare Group A and Group B.

Group A and group B are compared in every table

Table 2 and Table 6 should be consistent.

Tables 2 and 6 are joined in one Table (Table 3).

Where are Table 3 and Table 4?

Tables and Figures have been re-numbered.

Line 32-43: 9-16? Too many references for this expression. Suggest being more specific.

Irrelevant references have been deleted

Conclusions. Please revise the conclusion. It is too long. You should focus on the main finding of this study.

The conclusion section has been written again. We believe now that is more specific focusing on the main finding of this study.

Comparison between old and new version

Old version

New version

Introduction

The introduction has been rewritten

Hypothesis

Hypothesis and purpose of the research work are described at the end of the introduction section

Figures

Resolution of figures have been modified and improved

Table 2 has been deleted.

Data of table 2 has been added to table 3

References’ numbers

References numbering within the text have been modified. Irrelevant references have been deleted

In table 1 , has been replaced by .

Minor corrections (like. and ,) have been included to the text.

Paragraph 2.2.2 feed cost

The text in paragraph has been written again and cost has been re-calculated

Table’s numbers

Tables’s numbering within the text has been modified. There are no missing tables

Paragraph 4.2

Small changes within the text of this paragraph regarding % contains of solids and moisture

Paragraph 4.3

7 days old  broiler chickens

Paragraph 4.3.

Information about caloric value for group A and group B have been added

Table 7

Table 7 has been re-numbered to table 4 and now is more detailed

Section 4.4

The samples were homogenized from all broilers. 5 samples were written by mistake.

Conclusion

Conclusion has been revised

Discussion

Discussion has been revised in a previous stage

Reference list

Reference list has been revised and renumbered. Irrelevant references have been deleted

THE NOVELTY OF THE PAPER

The paper is novel and its novelty is based to the following points:

  • It is for the first time to combine pomegranate solid waste with avocado solid waste and produce silage
  • The inclusion of the solid wastes in the silage is very high creating substantial environmental benefit
  • The use of the silage by inclusion in the broilers nutrition led to the production of a bio-functional feed improving the broilers welfare and consequently to the production of clearly bio-functional meat with very low TBARS (marking non oxidized fat) and high ω-3 fatty acid content as well as high percentage of mono saturated fatty acids and low ratio of ω-6/ω-3 fatty acids.
  • The very important for the broiler breeders feeding performance ratio of the cost of feed/Kg of meat production was found to be much lower concerning the feed fortified with the bioactive silage compared to the conventional feed. This has significant positive economic implications regarding the use of the novel silage in broiler nutrition.
  • The feeding of the broilers with feed fortified with the novel silage is reducing by far the carbon footprint of this activity.

Reviewer 3 Report

The authors treated  the experimental well.

The number of chicken of the two groups is barely sufficient

The authors need to better explain the basic composition of the ration of the two groups before adding  pulp and seed wastes.

I ask the authors if they have made a histological investigation on the mucosa and intestinal villi of the chickens of two groups.

Author Response

COVER LETTER

Dear reviewer,

First of all, we would like to thank you for the efforts you made to improve this manuscript. We made all the proposed corrections. Please check our responses in the observations.

REVIEWER  3

Reviewer 3 Comments

Corrections

The number of chickens of the two groups is barely sufficient.

We agree with the reviewer but the trial performed inside university’s facilities and for this reason there was a limitation about the number of broilers. Furthermore, there are also other published studies with similar sample’s number.

Example 1. The research work with title “Feed supplemented with byproducts from olive oil mill wastewater processing increases antioxidant capacity in broiler chickens” published in Food and Chemical Toxicology 82 (2015) 42–49 mention that: “chickens were randomly divided into three experimental groups (12 broilers per group) as follows: (a) broilers fed with standard ration (control group), (b) broilers fed with ration containing OMWW ceramic microfiltration permeate (OMWW permeate group), and (c) broilers fed with ration containing OMWW ceramic microfiltration retentate (OMWW retentate group)”.

Example 2. The research work with title “Novel feed including bioactive compounds from winery wastes improved broilers' redox status in blood and tissues of vital organs” published in Food and Chemical Toxicology 102 (2017) 24-31 mention that: “Thirty female broilers (Hubbard) 2 days old were purchased from the ‘Agrafiotis’ aviary’, (Tyrnavos, Greece)”.

Example 3. The research work with title “Studies on the growth performance of different broiler strains at high altitude and evaluation of probiotic effect on their survivability” published in Scientific Reportsvolume 7, Article number: 46074 (2017) mention that: “For this study, a total of 50 no. one week old chicks from each breed of broiler strains, i.e. Vencobb, RIR cross-bred, and Hubbard brought from a commercial hatchery of plain areas were randomly selected. Thereafter, they were divided into 3 groups of 5 replicates with 10 chicks in each replicate according to completely randomized design so that mean values differ as little as possible”.

Example 4 The research work with title “Response of broiler chickens to different dietary crude protein and feeding regimens” published in Brazilian Journal of Poultry Science 7 (3) Sept 2005, mention that “There were four (4) treatment groups, each with three (3) replicates of ten (10) birds per replicate (One hundred and twenty (120) day-old Anak broilers in total)”.

Example 5. The research work with title “Experimental ochratoxicosis in broiler chickens” published in Avian Pathol. 2006 Aug;35(4):263-9 mention that “80 chicks that were fed a diet containing 0 parts per billion (ppb) (control, group 1), 400 ppb (group 2) or 800 ppb (group 3) OTA”

Example 6. The research work with title “The study of aloe vera powder effect as feed additive on the performance of broiler” published in IOP Conf. Series: Earth and Environmental Science 788 (2021) mention that: “total of 80 one-day-old broiler chicks (Cobb) were divided into four treatment groups with four replicates (5 chicks each) based on a completely randomized design”.

Example 7. Regarding other farm animals like piglets the research work with title “Feed supplemented with polyphenolic byproduct from olive mill wastewater processing improves the redox status in blood and tissues of piglets” published in Food and Chemical Toxicology 86 (2015) 319–327 mention that: “Thirty piglets were used that were from the pigsty of the Technical Education Institute of Thessaly. All piglets came from Landrace (mother) x Large White - Duroc - Pietrain (father) cross”.

Example 8. The research work with title “Effects of dietary partly destoned exhausted olive cake supplementation on performance, carcass characteristics and meat quality of growing lambs” published in Small Ruminant Research 156 (2017) 33–41 mention that: “Thirty two (32) weaned male lambs of the Greek breed Florina (Pelagonia) (65 ± 5 days of age) were used in this study”.

Example 9. The research work with title “Effects of Feeding Multinutrient Blocks Including Avocado Pulp and Peels to Dairy Goats on Feed Intake and Milk Yield and Composition” published in Animals 2020, 10, 194 mention that: “Twelve Murciano-Granadina dairy goats in the middle of the first lactation were selected and divided into 2 homogeneous groups of 6 goats each based on body weight (48.4 ± 2.40 kg), average voluntary feed intake (66.7 g dry matter (DM)/kg BW0.75), and milk yield (790 g milk/d) at the beginning of the experiment”.

The authors need to better explain the basic composition of the ration of the two groups before adding pulp and seed wastes.

Dear reviewer thanks for your observation the ingredients and the nutrient composition of basal feed (Group B) and on corn silage supplemented with pomegranate and avocado pulp and seed wastes (Group A) presented in table 4.

I ask the authors if they have made a histological investigation on the mucosa and intestinal villi of the chickens of two groups.

The purpose of the research work was not the examination of the microflora and bird health but explore the benefits of the antioxidant bioactive substances presented in both tested fruit wastes used as feed supplements on broiler’s meat. Moreover, there was no loss of broilers during the experiments from both groups. Furthermore, oxidative stress is higher when a broiler has not good welfare. Thus, there are data presented in the paper that give information about the oxidative status of the animals of the control group and of bioactive feed group expressed as measurement of the TBARS. There are also data on the fatty acid composition in the fat of the two respective groups and this are significant markers to prove less oxidative stress in the broiler group fed with the waste silage compared to the control group fed by conventional feed.

Comparison between old and new version

Old version

New version

Introduction

The introduction has been rewritten

Hypothesis

Hypothesis and purpose of the research work are described at the end of the introduction section

Figures

Resolution of figures have been modified and improved

Table 2 has been deleted.

Data of table 2 has been added to table 3

References’ numbers

References numbering within the text have been modified. Irrelevant references have been deleted

In table 1 , has been replaced by .

Minor corrections (like. and ,) have been included to the text.

Paragraph 2.2.2 feed cost

The text in paragraph has been written again and cost has been re-calculated

Table’s numbers

Tables’s numbering within the text has been modified. There are no missing tables

Paragraph 4.2

Small changes within the text of this paragraph regarding % contains of solids and moisture

Paragraph 4.3

7 days old  broiler chickens

Paragraph 4.3.

Information about caloric value for group A and group B have been added

Table 7

Table 7 has been re-numbered to table 4 and now is more detailed

Section 4.4

The samples were homogenized from all broilers. 5 samples were written by mistake.

Conclusion

Conclusion has been revised

Discussion

Discussion has been revised in a previous stage

Reference list

Reference list has been revised and renumbered. Irrelevant references have been deleted

THE NOVELTY OF THE PAPER

The paper is novel and its novelty is based to the following points:

  • It is for the first time to combine pomegranate solid waste with avocado solid waste and produce silage
  • The inclusion of the solid wastes in the silage is very high creating substantial environmental benefit
  • The use of the silage by inclusion in the broilers nutrition led to the production of a bio-functional feed improving the broilers welfare and consequently to the production of clearly bio-functional meat with very low TBARS (marking non oxidized fat) and high ω-3 fatty acid content as well as high percentage of mono saturated fatty acids and low ratio of ω-6/ω-3 fatty acids.
  • The very important for the broiler breeders feeding performance ratio of the cost of feed/Kg of meat production was found to be much lower concerning the feed fortified with the bioactive silage compared to the conventional feed. This have significant positive economic implication regarding the use of the novel silage in broiler nutrition.
  • The feeding of the broilers with feed fortified with the novel silage is reducing by far the carbon footprint of this activity.

Reviewer 4 Report

The paper is very interesting, and its topic corresponds to the scope of Molecules. However, it requires substantial revision.

Title:

- As there is an economic aspect to the work, perhaps this should be included in the title

Introduction:

- It is not correlated with aim of the study in Abstract (line 77-87)

-Line 88-91 transfer to chapter „Conclusion”

Results:

- Line 117:”.. and avocado by-products add (group A)…” and use abbreviations in the rest of the text. Harmonise throughout the text.  

- Line 131, 132-133: delete „(that includes broilers fed with ration supplemented with pulp and seeds of pomegranate and avocado by-products) and „(fed with basal ration-control)

Line 183-188: Delete „Comparing fat contents in breast of the two different groups, it was observed that fat content in group A (fed with ration supplemented with pulp and seeds of pomegranate and avocado by-products) was 1.99 slightly higher than fat content (1.79) observed in breast meat tissues in broilers fed with basal ration (group B). Moreover, similar results were observed for fat content in chicken’s leg meat (5.19 and 5.09 respectively).” repetition of the table results

- Line 188- 192: „Regarding protein content, in both groups, protein content found in breast was higher than protein content found in chicken’s leg, varied from 22.57-22.86 and 19.76-20.04 respectively. Protein content in leg meat of broilers fed with pulp and seeds of pomegranate and avocado by-products was slightly higher (20.04) than those observed in broilers of group B (19.76).” - as above. Please write, for example, how much it has increased.

- Line 194-196: „Finally, collagen in breast meat of broilers of group A was lower (0.79) than collagen content (0.88) found in breast meat of chickens fed with a basal content (group B). However, collagen content was slightly higher (1.07) in leg meat in group A comparing with collagen found in leg meat of group B (1.06).” – as above

- Section „2.2.4 Lipid and fatty acid profile” to beto be redrafted

- Line 209: full name then abbreviation

- Line 252-257: „According to Naht Dinh [49], palmitic and oleic acid are the most important FAs, not only in subcutaneous fat, but also in important muscles such as the Longissimus lumborum et thoracis. Since, linoleic and -linolenic are the main -6 and -3 PUFAs, respectively, and are important for human nutrition and health [50], their increase in chicken meat reveals the beneficial effect of inclusion of avocado and pomegranate addition in the feed in poultry diet on broiler’s quality. PUFA/SFA and (MUFA+ PUFA)/SFA ratios did not differ between treatments.” Moved to discussion

Discussion:

- The authors devote too much attention to the description of the results of other research but interpret the obtained results insufficiently. There is no exact interpretation of the results, which is necessary to see the publication as valuable.  Even where the authors try to explain their results, they actually repeat information contained in the literature to which they refer. There is no broader view of the problem. Therefore, I believe that this section is too modest and requires revision.

Material and methods:

- Please provide the number of Ethics Committee approval to conduct the research

- Correct and complete description; nevertheless, it should be describe how cockerels were selected within a group  (line 473)

- 25 chickens in a group? Not too few? Not very reliable results with such numbers

Conclusion:

- Line 558-559: „This meat is rich in ω3 fatty acids, mono-saturated fatty acids and presents very low and desirable ω6/ω3 ratio „repetition of the sentence above Line 551-552: „… very rich in mono-saturated fatty acids as well as, ω-3 fatty acids and in addition with reduced ratio of ω6/ω3.”

Author Response

COVER LETTER

Dear reviewer,

First of all, we would like to thank you for the efforts you made to improve this manuscript. We made all the proposed corrections. Please check our responses in the observations.

REVIEWER  4

Reviewer 3 Comments

Corrections

As there is an economic aspect to the work, perhaps this should be included in the title.

Dear reviewer thanks you very much for the observation but we believe that the current title describes better the research work.

It is not correlated with aim of the study in Abstract (line 77-87)

The introduction has been rewritten according to suggestions made by the previous reviewers (in previous evaluation) and now we believe that is more specific and not too long describing well the purpose of the research work.

Line 88-91 transfer to chapter „Conclusion”

It has been transferred to conclusion section as it is suggested.

Results: - Line 117:”.. and avocado by-products add (group A)…” and use abbreviations in the rest of the text. Harmonise throughout the text.  

The proposed addition in line 117 done.

Line 131, 132-133: delete „(that includes broilers fed with ration supplemented with pulp and seeds of pomegranate and avocado by-products) and „(fed with basal ration-control)

The two phrases have been deleted.

Line 183-188: Delete „Comparing fat contents in breast of the two different groups, it was observed that fat content in group A (fed with ration supplemented with pulp and seeds of pomegranate and avocado by-products) was 1.99 slightly higher than fat content (1.79) observed in breast meat tissues in broilers fed with basal ration (group B). Moreover, similar results were observed for fat content in chicken’s leg meat (5.19 and 5.09 respectively).” repetition of the table results

The phrase has been deleted.

Line 188-192: „Regarding protein content, in both groups, protein content found in breast was higher than protein content found in chicken’s leg, varied from 22.57-22.86 and 19.76-20.04 respectively. Protein content in leg meat of broilers fed with pulp and seeds of pomegranate and avocado by-products was slightly higher (20.04) than those observed in broilers of group B (19.76).” - as above. Please write, for example, how much it has increased.

The phrase has been rewritten as it is suggested

Line 194-196: „Finally, collagen in breast meat of broilers of group A was lower (0.79) than collagen content (0.88) found in breast meat of chickens fed with a basal content (group B). However, collagen content was slightly higher (1.07) in leg meat in group A comparing with collagen found in leg meat of group B (1.06).” – as above

The phrase has been rewritten as it is suggested

Section „2.2.4 Lipid and fatty acid profile” to beredrafted

The title of 2.2.4 section redrafted as indicated.

Line 209: full name then abbreviation

Full name is now included followed by abbreviation.

Line 252-257: „According to NahtDinh [49], palmitic and oleic acid are the most important FAs, not only in subcutaneous fat, but also in important muscles such as the Longissimus lumborum et thoracis. Since, linoleic and -linolenic are the main -6 and -3 PUFAs, respectively, and are important for human nutrition and health [50], their increase in chicken meat reveals the beneficial effect of inclusion of avocado and pomegranate addition in the feed in poultry diet on broiler’s quality. PUFA/SFA and (MUFA+ PUFA)/SFA ratios did not differ between treatments.” Moved to discussion.

The suggested phrase has been removed to the discussion section. The appropriate changes in references have been made.

Discussion: - The authors devote too much attention to the description of the results of other research but interpret the obtained results insufficiently. There is no exact interpretation of the results, which is necessary to see the publication as valuable.  Even where the authors try to explain their results, they actually repeat information contained in the literature to which they refer. There is no broader view of the problem. Therefore, I believe that this section is too modest and requires revision.

The discussion has been rewritten according to suggestions made by the previous reviewers (in previous evaluation).

Material and methods:- Please provide the number of Ethics Committee approval to conduct the research.

The procedure from the university of Thessaly Ethics committee is automated. Furthermore we examine plant derived by products as feed that are not risky to animal welfare.

- Correct and complete description; nevertheless, it should be describe how cockerels were selected within a group (line 473)

No cockerels involved in this study.Please check again the  line 473,

concrete mixer was used in order to mix the grounded corn and the 473 pomegranate-avocado by-products.”

- 25 chickens in a group? Not too few? Not very reliable results with such numbers

We agree with the reviewer but the trial performed inside university’s facilities and for this reason there was a limitation about the number of broilers. Furthermore, there are also other published studies with similar sample’s number.

Example 1. The research work with title “Feed supplemented with byproducts from olive oil mill wastewater processing increases antioxidant capacity in broiler chickens” published in Food and Chemical Toxicology 82 (2015) 42–49 mention that: “chickens were randomly divided into three experimental groups (12 broilers per group) as follows: (a) broilers fed with standard ration (control group), (b) broilers fed with ration containing OMWW ceramic microfiltration permeate (OMWW permeate group), and (c) broilers fed with ration containing OMWW ceramic microfiltration retentate (OMWW retentate group)”.

Example 2. The research work with title “Novel feed including bioactive compounds from winery wastes improved broilers' redox status in blood and tissues of vital organs” published in Food and Chemical Toxicology 102 (2017) 24-31 mention that: “Thirty female broilers (Hubbard) 2 days old were purchased from the ‘Agrafiotis’ aviary’, (Tyrnavos, Greece)”.

Example 3. The research work with title “Studies on the growth performance of different broiler strains at high altitude and evaluation of probiotic effect on their survivability” published in Scientific Reportsvolume 7, Article number: 46074 (2017) mention that: “For this study, a total of 50 no. one week old chicks from each breed of broiler strains, i.e. Vencobb, RIR cross-bred, and Hubbard brought from a commercial hatchery of plain areas were randomly selected. Thereafter, they were divided into 3 groups of 5 replicates with 10 chicks in each replicate according to completely randomized design so that mean values differ as little as possible”.

Example 4 The research work with title “Response of broiler chickens to different dietary crude protein and feeding regimens” published in Brazilian Journal of Poultry Science 7 (3) Sept 2005, mention that “There were four (4) treatment groups, each with three (3) replicates of ten (10) birds per replicate (One hundred and twenty (120) day-old Anak broilers in total)”.

Example 5. The research work with title “Experimental ochratoxicosis in broiler chickens” published in Avian Pathol. 2006 Aug;35(4):263-9 mention that “80 chicks that were fed a diet containing 0 parts per billion (ppb) (control, group 1), 400 ppb (group 2) or 800 ppb (group 3) OTA”

Example 6. The research work with title “The study of aloe vera powder effect as feed additive on the performance of broiler” published in IOP Conf. Series: Earth and Environmental Science 788 (2021) mention that: “total of 80 one-day-old broiler chicks (Cobb) were divided into four treatment groups with four replicates (5 chicks each) based on a completely randomized design”.

Example 7. Regarding other farm animals like piglets the research work with title “Feed supplemented with polyphenolic byproduct from olive mill wastewater processing improves the redox status in blood and tissues of piglets” published in Food and Chemical Toxicology 86 (2015) 319–327 mention that: “Thirty piglets were used that were from the pigsty of the Technical Education Institute of Thessaly. All piglets came from Landrace (mother) x Large White - Duroc - Pietrain (father) cross”.

Example 8. The research work with title “Effects of dietary partly destoned exhausted olive cake supplementation on performance, carcass characteristics and meat quality of growing lambs” published in Small Ruminant Research 156 (2017) 33–41 mention that: “Thirty two (32) weaned male lambs of the Greek breed Florina (Pelagonia) (65 ± 5 days of age) were used in this study”.

Example 9. The research work with title “Effects of Feeding Multinutrient Blocks Including Avocado Pulp and Peels to Dairy Goats on Feed Intake and Milk Yield and Composition” published in Animals 2020, 10, 194 mention that: “Twelve Murciano-Granadina dairy goats in the middle of the first lactation were selected and divided into 2 homogeneous groups of 6 goats each based on body weight (48.4 ± 2.40 kg), average voluntary feed intake (66.7 g dry matter (DM)/kg BW0.75), and milk yield (790 g milk/d) at the beginning of the experiment”.

Conclusion:

- Line 558-559: „This meat is rich in ω3 fatty acids, mono-saturated fatty acids and presents very low and desirable ω6/ω3 ratio „repetition of the sentence above Line 551-552: „… very rich in mono-saturated fatty acids as well as, ω-3 fatty acids and in addition with reduced ratio of ω6/ω3.”

The duplicated phrase has been deleted (lines 558-559).

Comparison between old and new version

Old version

New version

Introduction

The introduction has been rewritten

Hypothesis

Hypothesis and purpose of the research work are described at the end of the introduction section

Figures

Resolution of figures have been modified and improved

Table 2 has been deleted.

Data of table 2 has been added to table 3

References’ numbers

References numbering within the text have been modified. Irrelevant references have been deleted

In table 1 , has been replaced by .

Minor corrections (like. and ,) have been included to the text.

Paragraph 2.2.2 feed cost

The text in paragraph has been written again and cost has been re-calculated

Table’s numbers

Tables’s numbering within the text has been modified. There are no missing tables

Paragraph 4.2

Small changes within the text of this paragraph regarding % contains of solids and moisture

Paragraph 4.3

7 days old  broiler chickens

Paragraph 4.3.

Information about caloric value for group A and group B have been added

Table 7

Table 7 has been re-numbered to table 4 and now is more detailed

Section 4.4

The samples were homogenized from all broilers. 5 samples were written by mistake.

Conclusion

Conclusion has been revised

Discussion

Discussion has been revised in a previous stage

Reference list

Reference list has been revised and renumbered. Irrelevant references have been deleted

THE NOVELTY OF THE PAPER

The paper is novel and its novelty is based to the following points:

  • It is for the first time to combine pomegranate solid waste with avocado solid waste and produce silage
  • The inclusion of the solid wastes in the silage is very high creating substantial environmental benefit
  • The use of the silage by inclusion in the broilers nutrition led to the production of a bio-functional feed improving the broilers welfare and consequently to the production of clearly bio-functional meat with very low TBARS (marking non oxidized fat) and high ω-3 fatty acid content as well as high percentage of mono saturated fatty acids and low ratio of ω-6/ω-3 fatty acids.
  • The very important for the broiler breeders feeding performance ratio of the cost of feed/Kg of meat production was found to be much lower concerning the feed fortified with the bioactive silage compared to the conventional feed. This has significant positive economic implications regarding the use of the novel silage in broiler nutrition.
  • The feeding of the broilers with feed fortified with the novel silage is reducing by far the carbon footprint of this activity.
